# Naphthenic Acids Aggregation: The Role of Salinity

Renato D. Cunha [1,2], Livia J. Ferreira [3], Ednilsom Orestes [3,4], Mauricio D. Coutinho-Neto [3], James M. de Almeida [5], Rogério M. Carvalho [6], Cleiton D. Maciel [7], Carles Curutchet [1,2] and Paula Homem-de-Mello [3,*]

[1]  Departament de Farmàcia i Tecnologia Farmacèutica, i Fisicoquímica, Facultat de Farmàcia i Ciències de l'Alimentació, Universitat de Barcelona (UB), 08028 Barcelona, Spain
[2]  Institut de Química Teòrica i Computacional (IQTCUB), Universitat de Barcelona (UB), 08028 Barcelona, Spain
[3]  Center of Natural Sciences and Humanities, Federal University of ABC, Santo André 09020-580, SP, Brazil
[4]  Departamento de Ciências Exatas, Escola de Engenharia Industrial Metalúrgica de Volta Redonda, Universidade Federal Fluminense, Volta Redonda 27255-125, RJ, Brazil
[5]  CNPEM, Ilum School of Science, Campinas 13087-548, SP, Brazil
[6]  Research and Development Center, Leopoldo A. M. de Mello, Petrobras, Rio de Janeiro 21949-900, RJ, Brazil
[7]  Federal Institute of São Paulo, São Paulo 08571-050, SP, Brazil
*   Correspondence: paula.mello@ufabc.edu.br

**Abstract:** Naphthenic Acids (NA) are important oil extraction subproducts. These chemical species are one of the leading causes of marine pollution and duct corrosion. For this reason, understanding the behavior of NAs in different saline conditions is one of the challenges in the oil industry. In this work, we simulated several naphthenic acid species and their mixtures, employing density functional theory calculations with the MST-IEFPCM continuum solvation model, to obtain the octanol–water partition coefficients, together with microsecond classical molecular dynamics. The latter consisted of pure water, low-salinity, and high-salinity environment simulations, to assess the stability of NAs aggregates and their sizes. The quantum calculations have shown that the longer chain acids are more hydrophobic, and the classical simulations corroborated: that the longer the chain, the higher the order of the aggregate. In addition, we observed that larger aggregates are stable at higher salinities for all the studied NAs. This can be one factor in the observed low-salinity-enhanced oil recovery, which is a complex phenomenon. The simulations also show that stabilizing the aggregates induced by the salinity involves a direct interplay of $Na^+$ cations with the carboxylic groups of the NAs inside the aggregates. In some cases, the ion/NA organization forms a membrane-like circular structural arrangement, especially for longer chain NAs.

**Keywords:** naphthenic acids; molecular dynamics; MST-IEFPCM; log $P$; aggregation; solvation; salinity

## 1. Introduction

Naphthenic Acids (NAs) are one of the components present in petroleum. Therefore, considering water separated from petroleum during a produced step (called produced water), the presence of naphthenic acids is a significant concern. Furthermore, the toxicity of these species can affect marine life [1,2] and is one of the causes of ducts and extraction machinery corrosion [1,3–6].

NAs are carboxylic acids with or without cyclic chains and are usually present in the oil in different amounts. However, in the petroleum industry, it is very common to include all kinds of acids and call them naphthenic [7]. Some Brazilian crude oils have a high Total Acid Number (TAN), and NAs are commonly found in the reserves [8,9]. Recently, Porto et al. characterized the NAs of the Brazilian oil reserve by chromatography, separating them by the number of carbons and saturation, represented by the Double Bond Number (DBE) plus the z number [10]. A similar composition was also observed in produced water from Norwegian offshore oil platforms [7]. However, considering OSPW (oil sand produced

water) from Canada, the naphthenic acid portion has a more complex composition, not limited to carboxylic acid species [11].

Computational studies of petroleum components are essential to determine the properties of these compounds in different conditions. They have focused on all the steps of petroleum extraction, from corrosion studies to propositions of depollution and evaluation of environmental effects [12,13]. Asphaltenes and wax are the most studied components of petroleum. In some cases, computational and experimental studies are combined to improve the understanding of the interactions and reactions that occur in those systems and improve petroleum flow assurance and production methods [14–21].

NA separation has been studied both by experimental and computational methods. Some behavior of the NA was studied computationally in the literature, using quantum and classical methods to reproduce the NA properties [22–26]. In a previous computational study [27], we employed Molecular Dynamics to evaluate the interaction of three different NA (all with 14 carbon atoms) with a micelle. Alicyclic NAs intercalate with surfactant enhancing the micelle absorption capacity.

In the present work, we study the behavior of different NAs in aqueous environments with different saline concentrations. We aim to quantify the partition coefficient and the aggregation properties, employing quantum chemical solvation models and molecular dynamics (MD) simulations. Several simulations were performed for the primary six types of NAs present in Brazilian petroleum, as shown by a previous chromatographic study [10]. This study aims to characterize the impact of marine salinity conditions on the aggregation tendencies of different NAs and rationalize these properties regarding the NAs' hydrophobicity. This understanding will be valuable to develop technologies able to clean marine water near petroleum extraction spots polluted with these species and mitigate the corrosion of extraction machinery induced by them. Moreover, we can relate the NAs behavior to low-salinity-enhanced oil recovery mechanisms, as the acidic components of the oil could be released and dragged for extraction.

In the present contribution, we discuss the results of quantum chemical continuum solvation calculations aimed at assessing the hydrophobicity of several NA species through estimates of their octanol–water partition coefficients. The aggregation propensities of the NAs are obtained from classical MD simulations at different salinity conditions. Together, both techniques showed that longer chain acids are less hydrophilic and have more tendency to form higher-order aggregates. In addition to pure water, high and low-salinity water was simulated. We could show that larger aggregates are more stable at higher salinities, whereas the stable aggregates in low-salinity and pure water were smaller. These findings can be related to low-salinity-enhanced oil recovery strategies.

## 2. Materials and Methods

### 2.1. Quantum Chemical Continuum Solvent Calculations

In this work, we study a set of NAs compatible with the compositions observed in previous work considering the Brazilian scenario [10]. Based on gas chromatography coupled to mass spectrometry, this study demonstrated that the most common acids in crude oil are those shown in Figure 1. In such NAs, the carboxylic acid can be bonded to the ring or in the final part of the aliphatic chain; in the present study, we focus on the former, as we aim to assess the impact of changing salinity conditions and chain length/hydrophobicity on the aggregation properties of NAs.

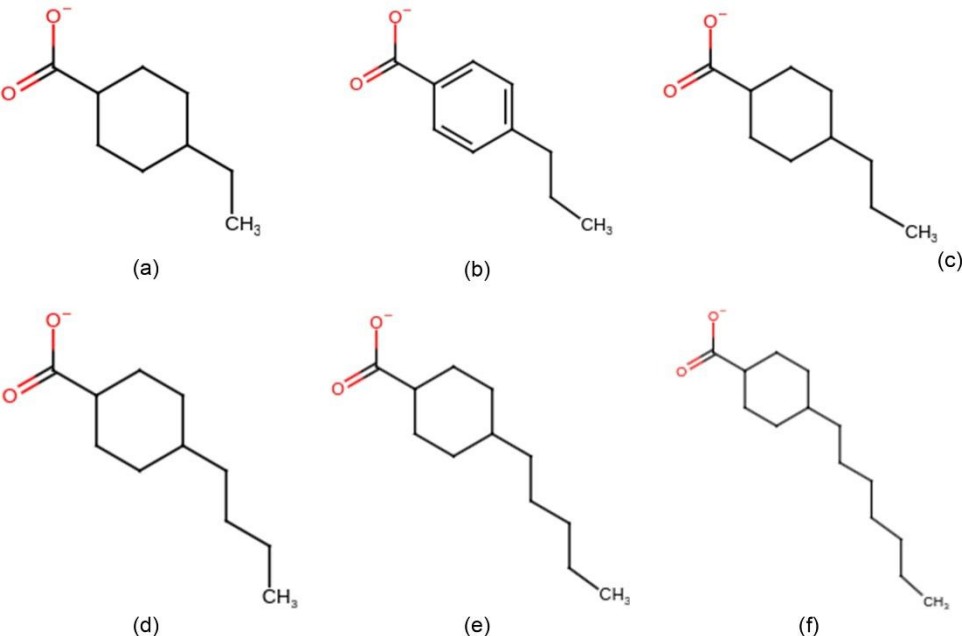

**Figure 1.** Molecular structure of the studied naphthenic acids: (**a**) c9, (**b**) c10 aromatic, (**c**) c10, (**d**) c11, (**e**) c12 and (**f**) c14.

The hydrophobicity of the NAs under study was explored using the MST-IEFPCM continuum solvation model developed in Barcelona [28,29]. In particular, the hydrophobicity was computed based on the octanol–water partition coefficient of the NAs, the log *P*. The MST-IEFPCM model has recently been assessed for the prediction of log *P* in the Statistical Assessment of Modeling of Proteins and Ligands (SAMPL) series of blind predictive challenges. In particular, the MST model showed excellent accuracy compared to other physics-based methods in SAMPL6 and SAMPL7 [30,31].

To estimate the partition coefficients, first, all geometries were optimized in water and n-octanol at the B3LYP/6-31G(d)-IEFPCM-MST level of theory, as implemented in Gaussian16 [32]. Next, all the minima were verified by vibrational frequency calculations. Then, single-point calculations were performed in a vacuum to estimate the solvation free energy in water and n-octanol, both for the neutral and the ionic species. Finally, from these solvation free energies, the partition coefficients were computed using the following relation (Equation (1)) with the transfer of free energy among the two solvents. The distribution coefficient (log *D*) was calculated based on the log *P* of the neutral species (log $P_N$) and of the ionic species (log $P_I$), as shown in Equation (2), where the $\delta = pH - pK_a$ for an acid [33]. We adopted a pKa of 4.9, as determined experimentally for similar naphthenic acids [34].

$$\log P = -\frac{\Delta\Delta G_{solv}(wat \rightarrow oct)}{2.303RT} = -\frac{\Delta G_{solv}(oct) - \Delta G_{solv}(wat)}{2.303RT} \tag{1}$$

$$\log D = \log\left(P_N + P_I \cdot 10^{\delta}\right) - \log\left(1 + 10^{\delta}\right) \tag{2}$$

### 2.2. Molecular Dynamics Simulations

We performed MD simulations for each NA, as shown in Figure 1, and for a mixture of NAs. Each system was prepared by solvating 60 NA molecules on an octahedral box. For the mixture, 10 NAs of each type were included, giving a total of 60 NAs. The simulations for each system were performed under three conditions, including pure water or saline solutions. We chose the following values of NaCl concentrations to screen different possibilities of salinity:

- 35 g/L—called here "low salinity" solution, comparable to injection water salinity; and
- 90 g/L—called here "high salinity" solution, comparable to the salinity of some pre-salt regions.

Thus, 21 systems were prepared, with the boxes having a volume of ~$1 \times 10^6$ Å$^3$, as shown for the NA mixture system in Figure 2. All boxes were constructed with Packmol [35]. We used the GAFF2 force field [36] and RESP atomic charges [37] obtained at the B3LYP/6-31G(d) level of theory to model the NA molecules. We also used the TIP3P water model and the ion parameters developed by Joung and Cheatham [38]. All simulations were prepared and run using the Amber 20 suite of programs [39].

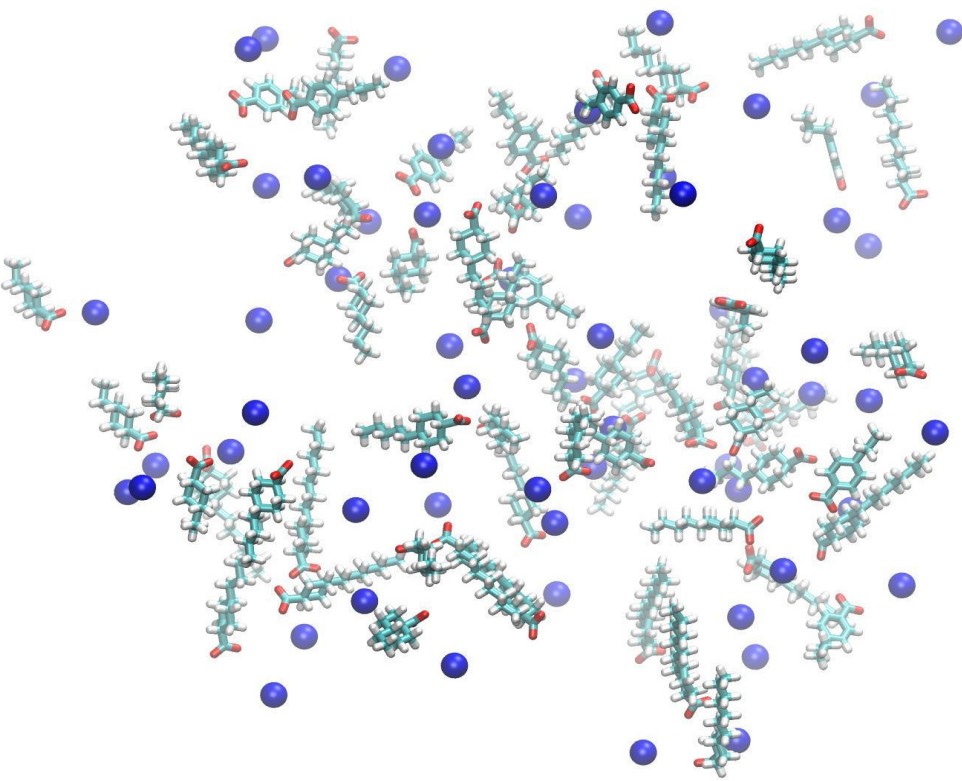

**Figure 2.** Example of an initial box of the NA mixture simulation with 60 NAs, 10 of each type (water molecules were omitted).

After an initial minimization, all systems were heated and equilibrated from 50 K to 300 K by running 50 ps NVT and 50 ps NPT simulations at 1 bar. Then, we extended the simulations for 1 μs with the NVT ensemble for production purposes. All simulations were performed using an integration time step of 4 fs by adopting the hydrogen mass repartitioning scheme developed by Roitberg and co-workers [40], together with the SHAKE algorithm to restrain all bonds involving hydrogen. In addition, we employed periodic boundary conditions and the particle-mesh Ewald approach to account for the long-range electrostatics and a non-bonded cutoff equal to 10 Å. Data were collected every 0.01 ps along the trajectories, and the analyses were made using Amber's cpptraj tool [41].

In order to analyze the data in detail, we wrote a Python code that calculates some key indicators to quantify aggregation properties along the MD trajectories based on the separation between the NAs. In the code, different NAs were included in the same aggregate whenever the distance between any two atoms of the NAs was lower than 4 Å. Then, for each frame of the trajectory, we calculated: (i) the percent of NAs aggregated and (ii) the number of NAs in the largest aggregate.

## 3. Results and Discussion

### 3.1. Hydrophobicity of NAs

The aggregation properties of organic molecules depend critically on their hydrophobic/hydrophilic properties, the critical quantity being the balance between intermolecular interactions established and the desolvation cost associated with the formation of aggregates. Therefore, a relevant physicochemical property associated with a molecule's intrinsic hydrophobic/hydrophilic character is its octanol–water partition coefficient, and the distribution coefficient, which takes into account all possible protonation states of the molecules in the water and organic phases. In the following, we present the results for the octanol–water partition and distribution coefficients as a function of chain length and aromaticity of NAs.

In Table 1, we report the results of the octanol–water partition coefficients of the neutral and ionic forms of the NAs (log $P_N$ and log $P_I$, respectively) and the corresponding distribution coefficients (log $D$) calculated for each NA molecule using the IEFPCM-MST continuum solvation model at the B3LYP/6-31G(d) level of theory. Negative log $P$/log $D$ values indicate a preference for the water phase compared to the organic phase for that molecule. In a previous study [27], we showed through explicit solvent simulations that the neutral protonated c14 presents an amphiphilic behavior in water-oil interfaces, with a greater preference for organic solvents. In this study, we are interested in aqueous solution phenomena, so all studied molecules were considered in their anionic deprotonated state in explicit solvent simulations, as will be shown in the next section. However, our continuum solvent calculations support our previous findings, with neutral c14 displaying a clear hydrophobic character with a log $P_N$ value of 4.81.

**Table 1.** Octanol–water partition coefficients (log $P$) for neutral (N) and ionic (I) species, and distribution coefficients (log $D$) of the naphthenic acids calculated using the IEFPCM-MST B3LYP/6-31G(d) continuum solvent model.

| NA Type | log $P_N$ | log $P_I$ | log $D$ (pH = 7) | log $D$ (pH = 8) |
|---|---|---|---|---|
| c9 | 2.26 | −2.14 | 0.16 | −0.82 |
| c10 aromatic | 2.41 | −2.02 | 0.31 | −0.67 |
| c10 | 2.78 | −1.62 | 0.68 | −0.30 |
| c11 | 3.26 | −1.16 | 1.16 | 0.18 |
| c12 | 3.81 | −0.63 | 1.71 | 0.73 |
| c14 | 4.81 | 0.41 | 2.71 | 1.73 |

Our results in Table 1 show the impact of the chain length on the NA partition properties. The overall effect is a net increase of the hydrophobicity by 2.5 log $P$ units when going from c9 to c14, a trend similar to the log $P$ of the neutral or ionic forms and maintained in the log $D$ values calculated at a neutral pH, or at the pH value of seawater, pH = 8. This trend arises from a variation of the water to octanol transfer free energies of −3.5 kcal/mol when going from c9 to c14, a change almost identical for neutral and anionic forms (data not shown). This variation arises from an increase in solvation free energies in water by +0.6 kcal/mol when going from c9 to c14, the longest-chain NA, and a corresponding decrease of −2.9 kcal/mol in n-octanol. Such variations can be explained given that hydration free energies decrease with increasing chain length due to attenuated electrostatic solute–solvent interactions. Still, this trend is reversed in n-octanol, as solvation energies increase due to increased dispersion-repulsion interactions with the solvent, given the stronger impact of non-electrostatic contributions in the organic phase. As can be seen, the aromatic character of the ring has a non-negligible effect on the hydrophobicity, as log $P$/log $D$ is increased by 0.4 in c10 compared to its aromatic analog.

Havre and co-authors [34] studied the oil–water interfacial behavior of some naphthenic acids, including partitioning the undissociated acid between the phases and the dissociation of carboxylic acid in the water phase, as the oil phase favors the protonated

species and water the unprotonated one. Our calculations are in excellent agreement with these experiments in terms of log $P$ values and the corresponding trend as a function of chain length. Indeed, upon addition of ~4–5 carbons, measured log $P_N$ values for NAs with one, two, or three rings increased 1.9, 2.6, and 2.3 units, respectively, with log $P_N$ values ranging from ~2.2 for 10 carbon NAs to ~4.1 for 14 carbon ones. These values are all very close to our results in Table 1, with a similar increase in log $P$ by 2.5 units when going from a system with 9 carbons to a system with 14 carbons.

*3.2. Aggregation Propensity of NAs*

In this section, we analyze the aggregation propensity of the NAs in different saline conditions, closely resembling those found in extraction spots at sea. To quantify the aggregation degree for the different NAs, we estimated different parameters, including the percent of NAs aggregated and the number of NAs in the largest aggregate.

In Figure 3, we report the evolution of the aggregation percentage along the MD trajectories for the six NAs under study at different saline concentrations, including pure water, at NaCl 35 g/L and NaCl 90 g/L. Our results indicate that both the chain length and salinity conditions strongly facilitate the aggregation of the NAs. In the simulations at NaCl 90 g/L, all NA types aggregate after 100–200 ns. If the salt concentration is reduced to NaCl 35 g/L, the c9 (Figure 3a) and c10 aromatic (Figure 3b) NAs only form dimers and trimers along the simulation. Still, they do not lead to larger aggregates, and the aggregation percent fluctuates between 20–30%, a trend explained by their higher hydrophilicity shown in Table 2. In contrast, all other NAs still lead to aggregates characterized by values over 90%. When the salt is missing in pure water, only the most hydrophobic NAs c11, c12, and c14 lead to aggregates in the simulated time.

The second property to facilitate the aggregation is related to the NAs composition. These acids have an amphiphilic behavior due to the alkyl chain and the negative charge on the carboxylic group. In pure water and low salinity solution, aggregation is more evident from c11 to longer acids. In Figure 4, we report the number of NAs in the largest aggregate (from a total of 60 NAs per box) along the trajectories. For c11, c12, and c14 (Figure 4d–f), the largest aggregate progressively includes almost all NAs in the simulation box. This trend is also consistent with the increased hydrophobicity of these compounds, as discussed in the previous section.

We now focus on the NA mixture, which contains all six types of NAs. In Figure 5, we report the evolution of the aggregation percent and the size of the largest aggregate along the MD trajectory for this system. The observed trends are similar to the simulations performed for single NA types, although we found more frequent transitions between aggregated and less compact states. Indeed, in pure water, the largest aggregate size reaches a value ~35 after 500 ns, but frequently fluctuates between this value and ~15–20 afterward. At NaCl 35 g/L and 90 g/L, the largest aggregate forms much faster, in ~100 ns. However, the maximum aggregate size still fluctuates along the trajectories between values ~50–60 and ~20–30, indicating that the largest aggregate often separates into two similar smaller aggregates.

**Table 2.** Percentage of aggregation for the NAs averaged over the last 500 ns of MD trajectory.

| NA Type | Water | Low Salinity Solution | High Salinity Solution |
|---|---|---|---|
| c9 | 19.5% | 29.4% | 93.0% |
| c10 aromatic | 21.0% | 29.7% | 85.0% |
| c10 | 32.7% | 93.5% | 98.1% |
| c11 | 81.3% | 98.5% | 99.4% |
| c12 | 94.1% | 99.6% | 100% |
| c14 | 99.4% | 100% | 100% |
| mixture | 66.3% | 91.1% | 97.2% |

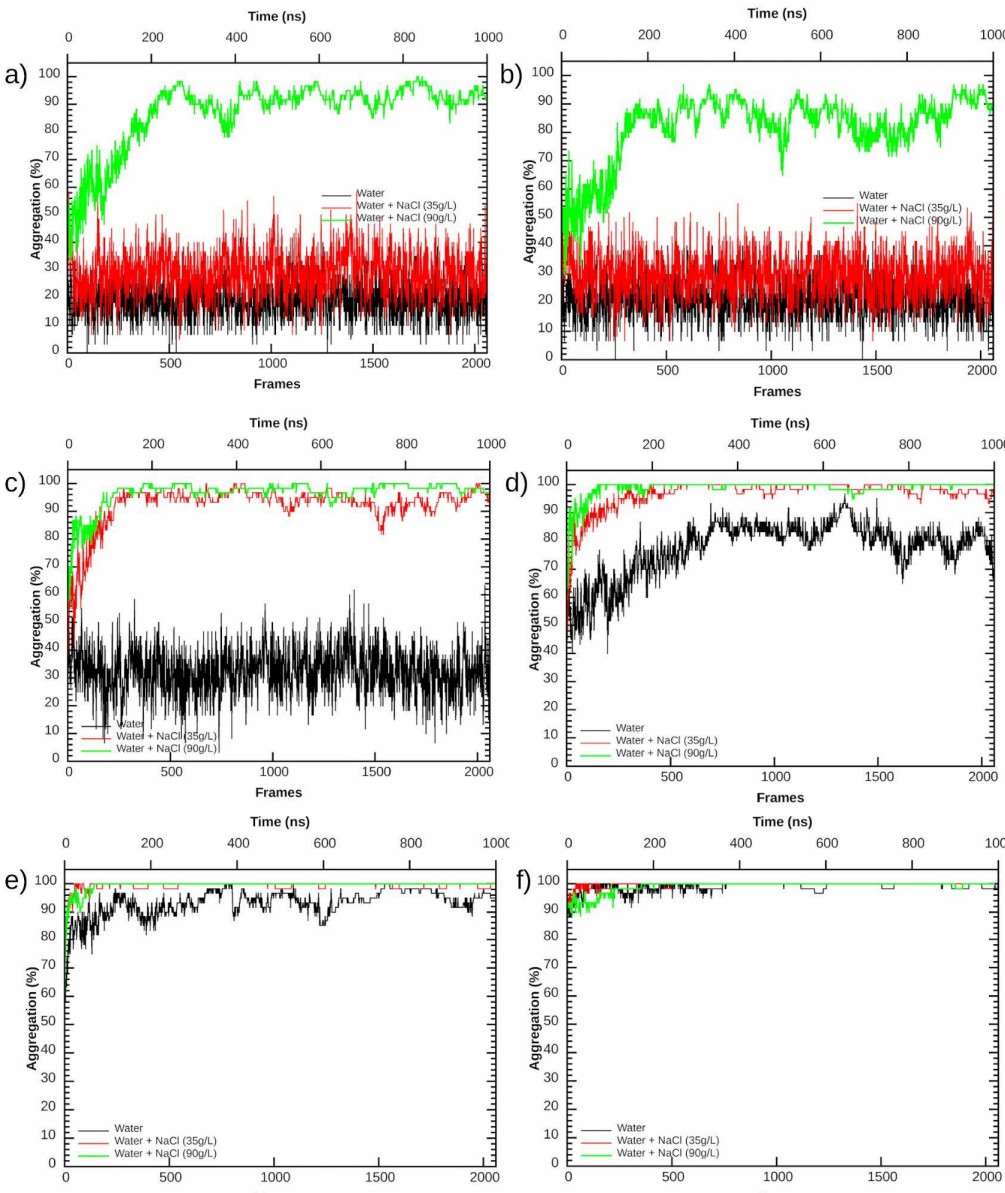

**Figure 3.** Percentage of aggregation along MD trajectories for different NAs in pure water (black), in low salinity solution (NaCl 35 g/L—red), and in high salinity solution (NaCl 90 g/L—green): (**a**) c9, (**b**) c10 aromatic, (**c**) c10, (**d**) c11, (**e**) c12 and (**f**) c14.

To quantify the final aggregation properties of the NAs, we averaged the aggregation percent along the final 500 ns structures sampled along the MD. The results are reported in Table 2, where lower values indicate that the acids can only form some dimers and trimers. On the other hand, large aggregation degrees describe the formation of higher order aggregates, with some molecules, however, still freely solvated in the box. Finally, aggregation degrees close to 100% indicate a total organization of the acids in the form of a single, large aggregate.

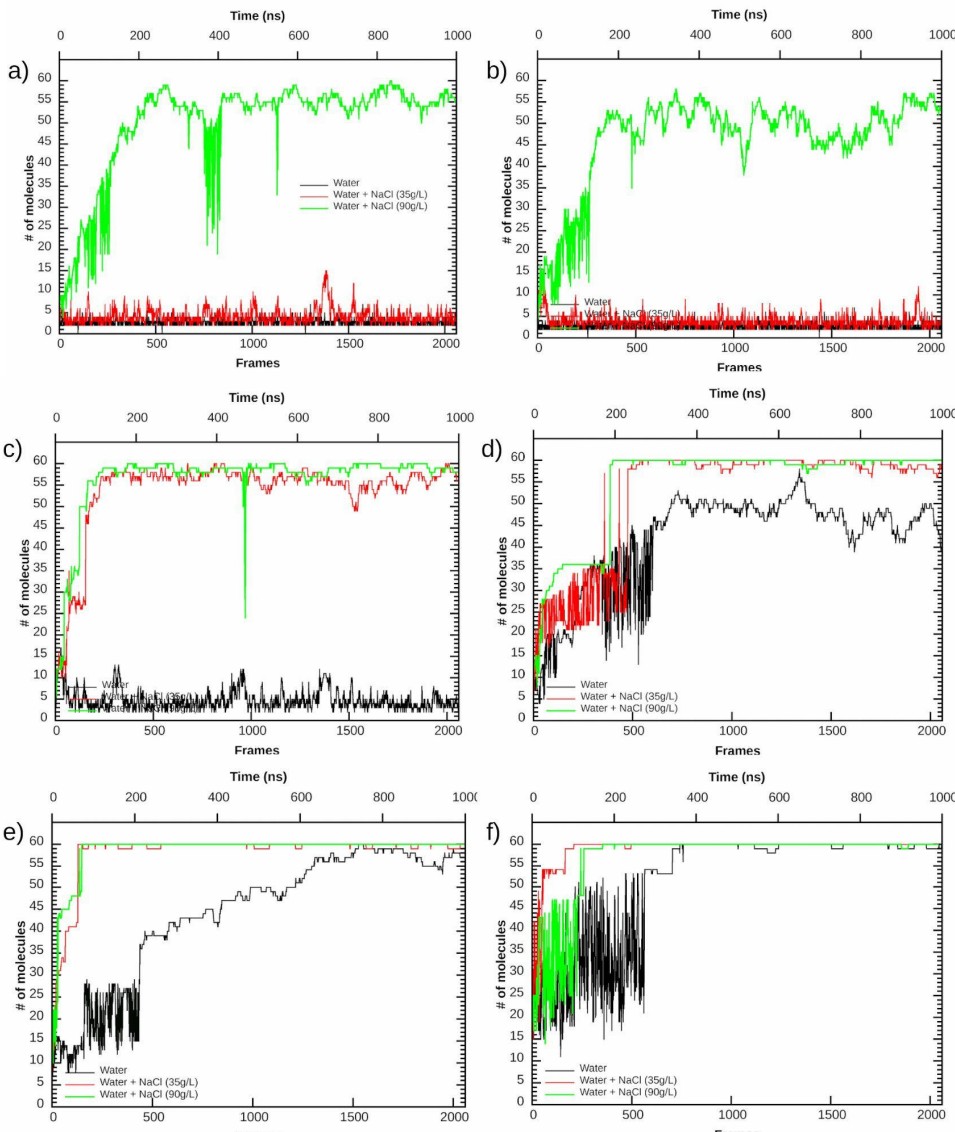

**Figure 4.** Size of largest aggregate along MD trajectories for different NAs in pure water (black), in low salinity solution (NaCl 35 g/L—red), and in high salinity solution (NaCl 90 g/L—green): (**a**) c9, (**b**) c10 aromatic, (**c**) c10, (**d**) c11, (**e**) c12 and (**f**) c14.

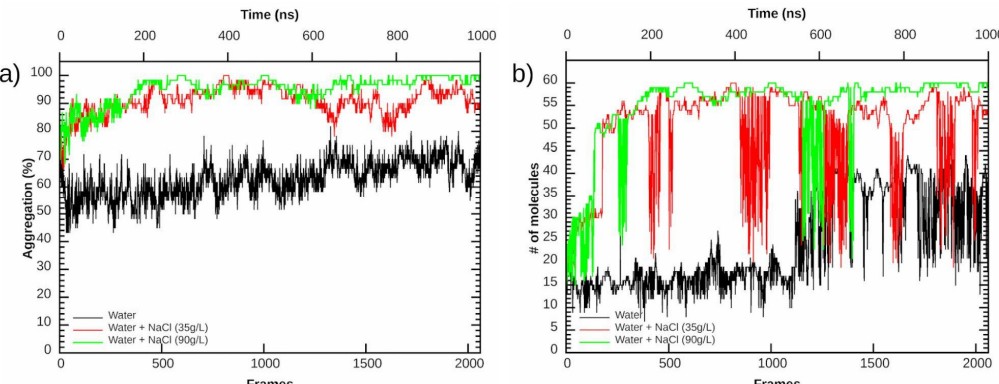

**Figure 5.** Percentage of aggregation (**a**) and size of largest aggregate (**b**) along the MD trajectory for the NA mixture system in pure water (black), in low salinity solution (NaCl 35 g/L—red), and in high salinity solution (NaCl 90 g/L—green).

It is interesting to analyze the different behavior between the c10 and c10 aromatic compounds. First, the alicyclic acid tends to aggregate more than the aromatic acid, which is somewhat unexpected, given that aromatic rings lead to strong $\pi$–$\pi$ stacking interactions, which should lead to an increased tendency to aggregate. However, the log $P_I$ of c10 is $-1.6$, whereas the c10 aromatic compound displays a log $P_I$ value of $-2.0$, as shown in Table 1. This can be explained by the formation of (weak) hydrogen bonds between aromatic rings and water [42]. In fact, these interactions are possible because the aromatic ring retains part of the negative charge of the molecule. As can be seen in Tables S1 and S2, the oxygen atoms for the c10 aromatic are slightly less negative than the oxygen atoms of the other NAs. This increased hydrophilicity thus seems to be the reason for the smaller tendency to form aggregates of the c10 aromatic system. Indeed, Table 2 shows that in pure water, the aggregation degree of c10 (33%) is larger than in c10 aromatic (21%). We note, however, that specific salt effects could also contribute to the increased aggregation behavior observed for c10 in the saline solutions.

The averaged aggregation degrees are consistent with the trends shown in Figures 3–5 along the MD simulations, confirming the impact of the alkyl chain size, and the salinity, as modulators for the aggregate formation. Table 2 shows that the two larger NAs, c12 and c14, aggregated in all salinity conditions and that the NaCl 90 g/L solution induces aggregation for all the compounds. To illustrate this behavior, in Figure 6, we show the last structure of the MD trajectories for the NA mixture in pure water, low salinity, and high salinity conditions. The growth of the aggregates is visible with the salinity of the environment. At high salt conditions, only one NA is outside the aggregate.

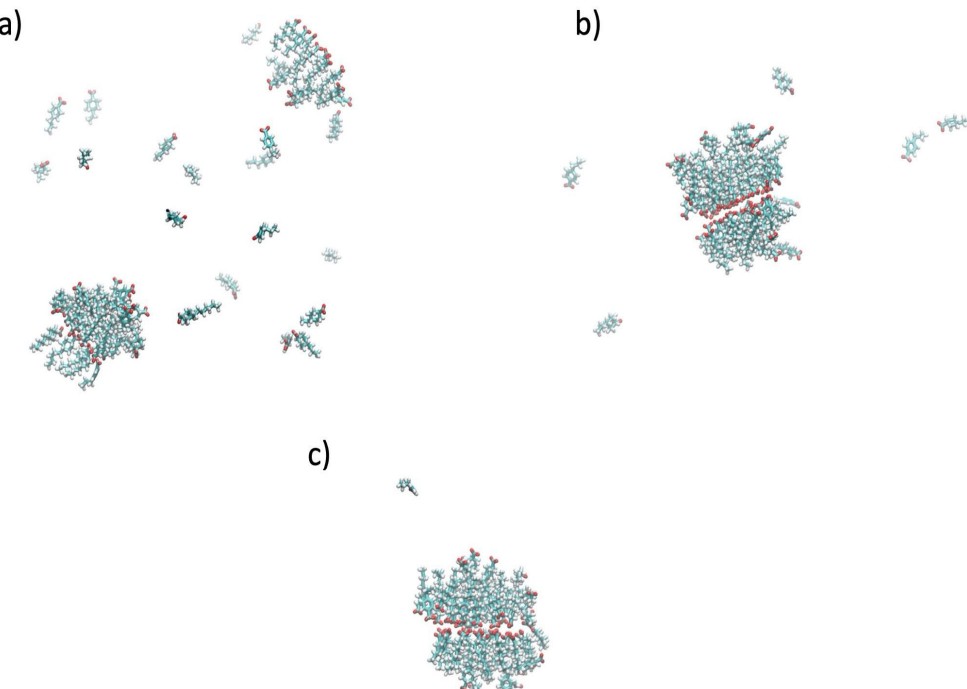

**Figure 6.** Structure of the NA mixture corresponding to the last frame of the MD trajectory: (**a**) pure water, (**b**) low salinity solution (NaCl 35 g/L), and (**c**) high salinity solution (NaCl 90 g/L).

Additionally, the largest aggregate size (Figure 5) shows that the aggregate is formed in the first 100 ns of the simulation. After this, other NA molecules enter progressively inside the aggregate. This result is interesting from the low-salinity-enhanced oil recovery point of view [43,44], as the formation water found in reservoirs is usually high-salinity [45]. The injection of low-salinity water would lower the salt concentration, destabilizing large aggregates, thus releasing part of the acid content that could be dragged for extraction.

In order to characterize the size of the aggregates, we have also computed radial distribution functions (RDF), quantifying the distribution of NAs around each single NA center of mass. The results, reported in Figure 7, show the density g(r) as a function of the distance between NAs' centers of mass. These densities indicate that the size of the aggregates is ~30 Å for fully aggregated states.

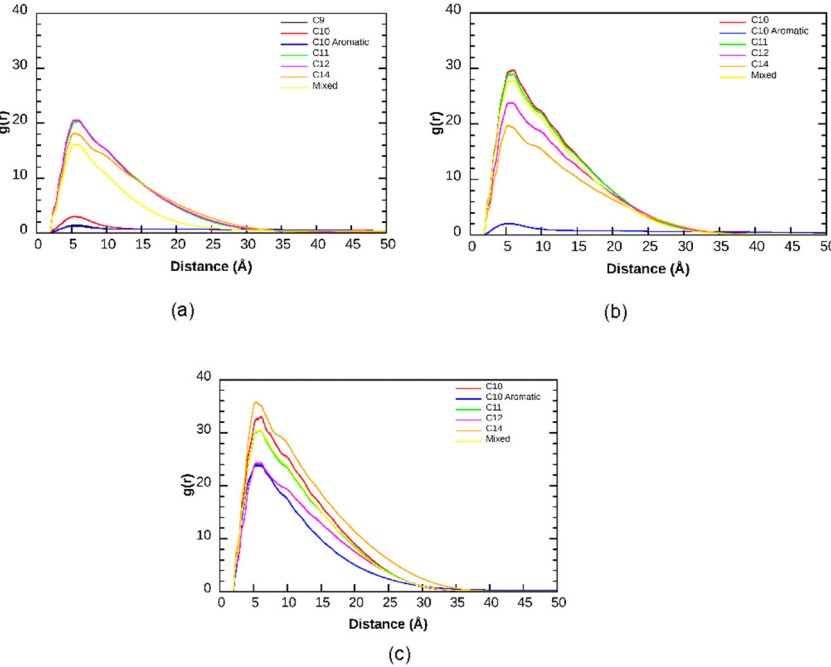

**Figure 7.** Radial distribution function quantifying the distribution of NAs around each single NA center of mass in (**a**) pure water, (**b**) low salinity solution (NaCl 35 g/L), and (**c**) high salinity solution (NaCl 90 g/L).

An interesting result of this analysis is the salinity's significant role in mediating the aggregates' formation. A detailed structural analysis of the aggregate morphology and the relative arrangements of NAs indicate that they adopt a structure that, at first sight, resembles a bilayer-like structure, as shown in Figure 8. However, the polar groups adopt a circular arrangement, with the center of the aggregate having an apolar character, as shown in the side view (Figure 8b). Interestingly, this bilayer-like circular arrangement displays the polar groups on the inner part of the aggregate, with a layer of sodium cations placed in the interior and interacting with the NA carboxylic groups. This can be seen in Figure 8, where Na$^+$ cations are arranged along the bilayer near the charged oxygens of the acids. Very similar behavior is found for all the systems studied when large aggregates are formed, regardless of the alkyl chain size. In order to shed light on the structure of the water along this arrangement, we show the radial distribution functions quantifying the distribution of the water's center of mass around the oxygen atoms of the NAs in Figure 9. As expected, the density of water diminishes as the aggregation degree increases, with longer alkyl chains and at high salinity conditions. However, it is interesting to observe that even in the most aggregated states, the first water solvation shell around the acid groups of the NAs is maintained with persistent and well-defined interactions, and some ordering can also be observed for the second solvation shell.

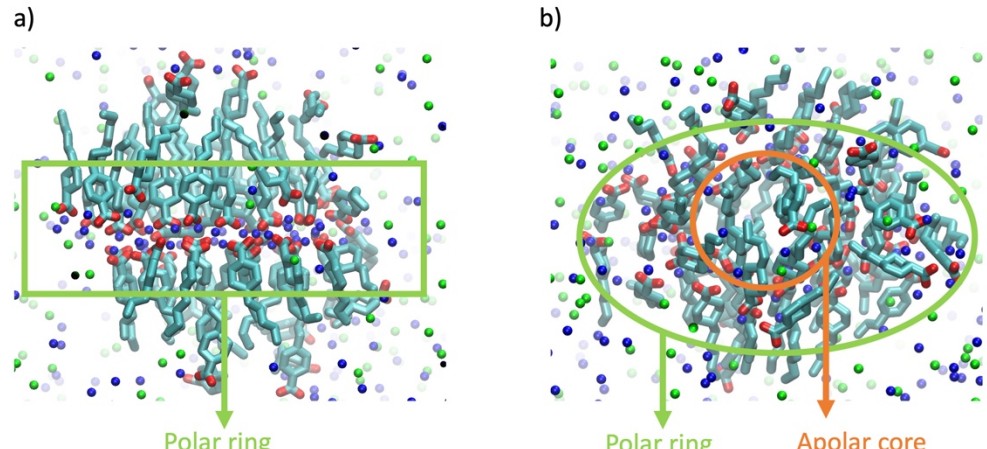

**Figure 8.** Structure of the aggregate found for the NA mixture at high salinity solution (NaCl 90 g/L). Blue spheres represent Na$^+$ ions, green spheres Cl$^-$ anions, and red and cyan spheres NA oxygen and carbon atoms. (**a**) Side view, (**b**) top view.

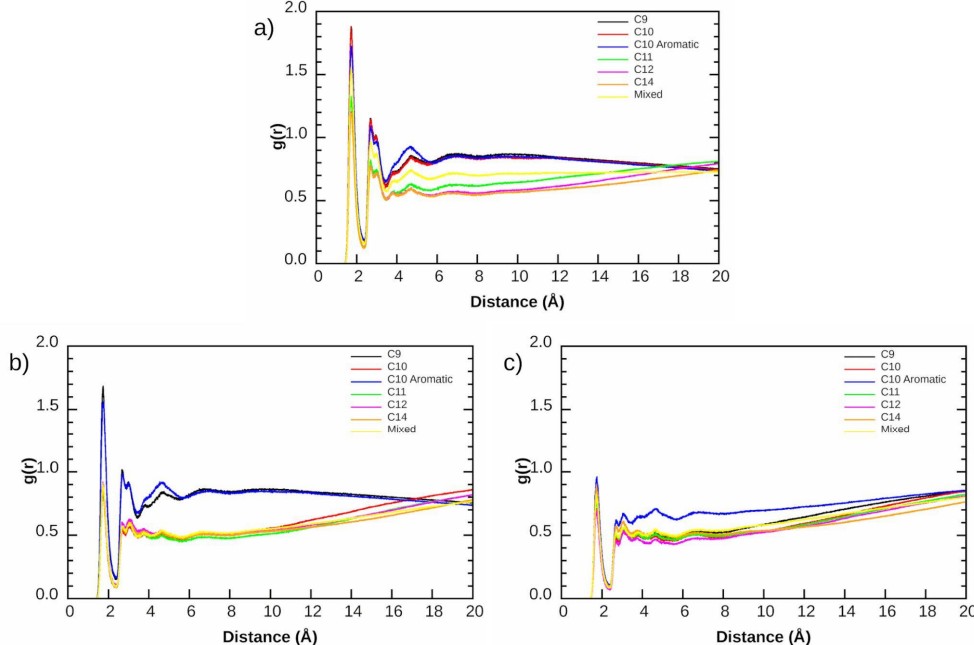

**Figure 9.** Radial distribution function quantifying the distribution of water (center of mass) around oxygen atoms of the NAs in (**a**) pure water, (**b**) low salinity solution (NaCl 35 g/L), and (**c**) high salinity solution (NaCl 90 g/L).

## 4. Conclusions

In the present work, we studied the behavior of different naphthenic acids in aqueous environments with varying saline concentrations. We aimed to quantify the partition coefficient and aggregation properties through quantum chemical solvation models and MD simulations. The results showed a clear relationship between (i) the size of the alkyl chain, which determines the hydrophobicity of the NAs as estimated by their octanol–water partition and distribution coefficients, and (ii) the salt concentration on the formation of large aggregates. For example, the NA with nine carbons (c9) only leads to small aggregates in the form of dimers or trimers. Still, the same compound on saline water (90 g/L) forms a large aggregate, with 93% of the NA molecules participating in this arrangement.

On the other hand, the NA with 14 carbons (c14) forms large aggregates in all cases, regardless of the salt conditions. The aggregation of the acids self-organizes in an interesting way: a layer of Na$^+$ ions organized between the oxygens of NA molecules, resembling a

circular membrane-like organization. The IEFPCM-MST calculations show a clear relation between the hydrophobicity of the NAs, as estimated by their log *P* and log *D* values, and the formation of aggregates. The most hydrophobic NAs can quickly form large aggregates compared to more hydrophilic NAs.

The present work improves our understanding of the structural behavior of NAs in aqueous environments, mimicking reservoir salinities. The low-salinity results showing smaller aggregate sizes can contribute to low-salinity-enhanced oil recovery, as the stable aggregates present in the higher salinity formation water can be partially dissolved when the salinity is lowered.

**Supplementary Materials:** The following supporting information can be downloaded at: https://www.mdpi.com/article/10.3390/computation10100170/s1, Table S1. RESP atomic charges calculated at B3LYP/6-31G(d) level of theory for the following naphthenic acids: c9, c10 aromatic, and c10; Table S2. RESP atomic charges calculated at B3LYP/6-31G(d) level of theory for the following naphthenic acids: c11, c12, and c14.

**Author Contributions:** Conceptualization, P.H.-d.-M., R.D.C., C.C. and R.M.C.; methodology, R.D.C., E.O., J.M.d.A., M.D.C.-N., C.D.M., C.C. and P.H.-d.-M.; software, R.D.C. and C.C.; formal analysis, R.D.C., L.J.F., C.C. and P.H.-d.-M.; resources, P.H.-d.-M. and C.C.; data curation, R.D.C. and L.J.F.; writing—original draft preparation, R.D.C., L.J.F., C.C. and P.H.-d.-M.; writing—review and editing, all authors; funding acquisition, C.C. and P.H.-d.-M. All authors have read and agreed to the published version of the manuscript.

**Funding:** This study was partially funded by the Coordenação de Aperfeiçoamento de Pessoal de Nível Superior—Brasil (CAPES)—Finance Code 001; R.D.C. acknowledges with thanks a PREDOCS-UB fellowship funded by the University of Barcelona (Ref. 2020-88). C.C. and R.D.C. acknowledge financial support from the State Research Agency/Spanish Ministry of Science and Innovation (AEI/10.13039/501100011033, grants MDM-2017-0767 and PID2020-115812GB-I00). P.H.M. thanks Conselho Nacional de Desenvolvimento Científico e Tecnológico (CNPq—#306585/2019-7) and Financiadora de Estudos e Projetos (FINEP—#0038/21) for the provided grants.

**Data Availability Statement:** Not applicable.

**Conflicts of Interest:** The authors declare no conflict of interest.

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
