# Peer review of "Naphthenic Acids Aggregation: The Role of Salinity"

_computation, doi:10.3390/computation10100170_

Round 1

Reviewer 1 Report

The manuscript presents an extensive set of calculations and in-depth analysis for a series of naphthenic acid molecules and their aggregates as a function of salinity.  DFT calculations at an appropriate level and with a solvent model are first used to estimate the water/octanol partition function for the molecules with various chain lengths coming to a not unexpected conclusion that the longer chains are more hydrophobic. Then MD simulations study the aggregation process directly, pointing out the trends as a function of chain length and for two different salt concentrations.  Overall the topic is of practical importance (acidity in the "oil patch") and the study yields insight into the complex aggregation process.  The paper would interest a wide cross section of Computation readers.

However, one aspect isn't clear to me and I think the manuscript should be revised to better explain, first, how the radii of gyration were calculated (which molecules are included in the aggregate and on what basis) and more details on the intriguing "twin" aggregate (called " bilayer") that is already apparent in Figure 5 and shown in more detail in Figure 7. Are all of the molecules included in the calculation of Rg? Wouldn't it be informative to have two values of Rg for each member of the "twin"? Could longer simulation times lead to a rearrangement with all of the hydrophobic tails interacting? Or is this unlikely? I think that a more extensive analysis/discussion of these questions could strengthen the paper.

Reviewer 2 Report

In this piece of research the authors have simulated several naphthenic acid (NA) species and mixtures to obtain the partition coefficients and to understand aggregation in pure water, low-salinity, and high-salinity environments. Their results could be relevant to petroleum extraction.

I believe this work deserves to be published in a top journal. However, there are two questions I would like the authors to answer.

1.  Can the authors analyze whether the NA molecules tend to bond similar molecules when forming aggregates in NA mixtures or, on the contrary, they are rather promiscuous and form their aggregates with no matter which NA? I am especially curious about the c10 aromatic species. In a mixture of NA's, do the c10 aromatic molecules tend to form clusters with other c10 aromatic molecules or aggregates are rather random mixtures of different NA's?

2. Can the authors discuss solvent (water) structure?

Reviewer 3 Report

This manuscript by Cunha et al. described a QM and MD study of the aggregation behavior of naphthenic acids (NA), and in particular the influence of salt concentration. The main finding was that high salt concentration promotes the aggregation of all the NAs studied in this work. This work is potentially publishable after the following issues are addressed.

1. The authors argued that c10 has a higher tendency for aggregation than c10 aromatic, and this trend correlates with the hydrophilicity as indicated by the water/octanol partition coefficient (Table 2, Line 244-248). However, c10 and c10 aromatic have similar aggregation behaviors in pure water, and their difference only exists in salt solutions, while the partition coefficient does not reflect the influence of salt. Therefore, the trend could be due to specific salt effects rather than hydrophilicity.
2. Since the aim is to study the influence of salinity, it will be more helpful to use a solvation model that includes salt concentration, such as the Generalized Born model (
https://pubmed.ncbi.nlm.nih.gov/11754341/).
3. In Line 182-183, the authors stated that "
Direct comparison with our computed results is not possible as one expects a complex protonation equilibrium that influences the NA partitioning..." The comparison is actually possible by including pKa, which can be obtained from experiments or from simple QM calculations (e.g. https://doi.org/10.1021/ja010534f).

4. Figure 3 and Figure 4. The authors used the radius of gyration (Rg) as a main metric for the aggregation. It seems that Rg was calculated by using the coordinate of all solute molecules. However, Rg is mainly used on polymers; the calculation on a periodic system of small solute molecules is not appropriate. For example, there are sharp pikes in Figures 3a and 3c, which are likely because a large aggregate crossed the boundary, making the Rg ill-defined. I would suggest using the number of aggregates or average aggregate size to monitor the time evolution of the aggregation behavior.
5. The authors should describe how the aggregation degree in Table 2 was calculated.

Round 2

Reviewer 1 Report

The authors have responded appropriately and adequately to previous comments.  I think the manuscript should now be published and I congratulate the authors for this interesting work!

Reviewer 3 Report

The authors have fully addressed my comments.